# Snow Parameters Inversion from Passive Microwave Remote Sensing Measurements by Deep Convolutional Neural Networks

**DOI:** 10.3390/s22134769

**Published:** 2022-06-24

**Authors:** Heming Yao, Yanming Zhang, Lijun Jiang, Hong Tat Ewe, Michael Ng

**Affiliations:** 1Department of Mathematics, The University of Hong Kong, Pokfulam Road, Hong Kong 999077, China; yaohmhk@hku.hk; 2Department of Electrical and Electronic Engineering, The University of Hong Kong, Pokfulam Road, Hong Kong 999077, China; ymzhangj@connect.hku.hk (Y.Z.); jianglj@hku.hk (L.J.); 3Department of Electrical and Electronic Engineering, Universiti Tunku Abdul Rahman, Perak 31900, Malaysia; eweht@utar.edu.my

**Keywords:** machine learning, deep convolutional neural networks (CNNs), passive microwave remote sensing (PMRS), inversion, dense medium radiative transfer (DMRT)

## Abstract

This paper proposes a novel inverse method based on the deep convolutional neural network (ConvNet) to extract snow’s layer thickness and temperature via passive microwave remote sensing (PMRS). The proposed ConvNet is trained using simulated data obtained through conventional computational electromagnetic methods. Compared with the traditional inverse method, the trained ConvNet can predict the result with higher accuracy. Besides, the proposed method has a strong tolerance for noise. The proposed ConvNet composes three pairs of convolutional and activation layers with one additional fully connected layer to realize regression, i.e., the inversion of snow parameters. The feasibility of the proposed method in learning the inversion of snow parameters is validated by numerical examples. The inversion results indicate that the correlation coefficient ($R^{2}$) ratio between the proposed ConvNet and conventional methods reaches 4.8, while the ratio for the root mean square error (RMSE) is only 0.18. Hence, the proposed method experiments with a novel path to improve the inversion of passive microwave remote sensing through deep learning approaches.

## 1. Introduction

As an important informative indicator for climate change, snowpack presents both the surface energy and water balance in a certain region [1,2]. Passive microwave remote sensing (PMRS) data have been widely employed to analyze snowpack because passive microwave remote sensing schemes can be applied in various weather and can penetrate clouds and snow [3,4]. Generally, the analysis and retrieval of snowpack by passive microwave measurements is based on the physical scattering model, which can produce both backscatter and brightness temperature measured from the physical parameters of the snowpack [5,6,7,8]. In this inversion process, multiple scattering in passive microwave remote sensing problems can be a dominant effect because the relation between remote sensing measurements and the medium parameters is highly nonlinear [5,6,7,8]. In the last decades, the research related to the inversion algorithm to obtain parameters of the snowpack by PMRS data has developed rapidly [9,10]. However, these conventional inversion methods, such as the conventional iterative method [11], Empirical formulas method [12] and artificial neural network (ANN) method [13,14], often make the process computationally expensive and even ill-posed [11,12,13,14,15].

The application of machine learning (ML) in advanced computational electromagnetics, such as object monitoring [16], electromagnetic simulation [17,18,19,20,21] and field-circuit cosimulations [22,23], was initiated a long time ago. Machine learning methods aim to derive the potential mapping disciplinarian by employing the same pattern’s training data and predicting the new output. Due to recent blooming learning technologies, the convolutional neural network (ConvNet) [24,25] has become one of the most significant methods in deep learning-based applications, such as imaging processes [24,25]. Various rough surface theory methods have been widely studied and applied in the simulation of snow [26,27]. However, the inverse problem still faces the challenge of nonlinearity and high computational complexity, which is still an open issue. In fact, machine learning methods such as conventional multiple layers artificial neural network (ANN) technique [14,28] and support vector machine [29], have been used to inverse parameters of the snowpack based on PMRS data. However, the inversion based on conventional machine learning methods suffers from limited accuracy, even with complex structure [21,28,29].

In this paper, we propose the employment of a deep ConvNet for inverting snow parameters (the thickness *t* and the temperature *T* of the snowpack) from passive microwave remote sensing measurements. The basic process is to use the input-output pairs generated by the scattering simulation model to train the proposed deep ConvNet. Once the ConvNet is trained, it can invert snow parameters (the thickness *t* and the temperature *T* of the snowpack) speedily and accurately from the measurements. The advantages of the proposed method are: (1) High accuracy: the proposed inverse ConvNet model can provide results with high accuracy, compared with support vector machine and conventional artificial neural networks trained for the inversion of snow parameters from passive microwave remote sensing [21,28,29]; (2) Simplicity: the training of the proposed deep ConvNet is merely based on the data simulated numerically using computational electromagnetic method, instead of using experimental measurement data; (3) High Noise Tolerance: the proposed ConvNet is of strong anti-interference and its accuracy is high, even though significant interference is added. Compared to the conventional artificial neural networks (ANNs) and the support vector machine (SVM) method [21,28,29], the proposed deep ConvNet can more efficiently map the relations between inputs and outputs, mainly by the convolutional layer and activation layer [24,25]. Consequently, the correlation coefficient ($R^{2}$) of the ConvNet method can be about 4.8 times as large as that of SVM for inversing *T*. Meanwhile, the root mean square error (RMSE) of the ConvNet method can be only 0.18 of the ANN method for inversing *t*. A specific comparison between different approaches is provided in Section 3.

The remainder of this article is organized as follows: In Section 2, the passive microwave remote sensing, the snowpack model, and dense media radiative transfer (DMRT) model formulation is briefly reviewed, followed by a description of the proposed deep ConvNet structure. Then, the training process of the proposed ConvNet for snow parameters inversion is described. In Section 3, numerical examples are provided to present the validity and precision of the proposed method, which are also compared with the results obtained by the conventional artificial neural network. Finally, the conclusion is given in Section 4.

## 2. Formulations

### 2.1. Snow Model and DMRT Model for PMRS

The model of the microwave emission behaviour for multilayer snowpacks is shown in Figure 1. It has been demonstrated that dense medium radiative transfer (DMRT) shows high validity and efficiency [30]. Thus, the input-output pairs for training ConvNet are generated by utilizing DMRT based on the quasicrystalline approximation (QCA) in this research. The DMRT formulas for PMRS could be simplified by amending these formulas for active microwave remote sensing because of the azimuthal symmetry. The standard formulation of the DMRT in the multilayer dielectric medium can be expressed as:(1)$$\begin{array}{cc} \; & \cos\theta \frac{d}{dz}I\left(\theta ,z\right)=-\kappa_{e}I\left(\theta ,z\right)+S\left(\theta ,z\right)+\kappa_{a}T\\ \; & -\cos\theta \frac{d}{dz}I\left(\pi -\theta ,z\right)=-\kappa_{e}I\left(\pi -\theta ,z\right)+W\left(\theta ,z\right)+\kappa_{a}T \end{array}$$
where
(2)$$\begin{array}{cc} \; & S\left(\theta ,z\right)=\int_{0}^{2\pi }d\theta^{\prime }\sin\theta^{\prime }\left[P\left(\theta ,\phi ;\theta^{\prime },\phi^{\prime }=0\right)\cdot I\left(\theta^{\prime },z\right)+\right.\\ \; & \left.P\left(\theta ,\phi ;\pi -\theta^{\prime },\phi^{\prime }=0\right)\cdot I\left(\pi -\theta^{\prime },z\right)\right]\\ \; & W\left(\theta ,z\right)=\int_{0}^{2\pi }d\theta^{\prime }\sin\theta^{\prime }\left[P\left(\pi -\theta ,\phi ;\theta^{\prime },\phi^{\prime }=0\right)\cdot I\left(\theta^{\prime },z\right)+\right.\\ \; & \left.P\left(\pi -\theta ,\phi ;\pi -\theta^{\prime },\phi^{\prime }=0\right)\cdot I\left(\pi -\theta^{\prime },z\right)\right] \end{array}$$

Herein, I(θ,z) denotes the specific intensity of horizontal and vertical polarizations, which is independent of the azimuthal angle ϕ. $\kappa_{e}$ and $\kappa_{a}$ are the extinction coefficient and absorption coefficient respectively. The phase matrix P is simplified by using 1–2 frames [28,31]. T means the transmissivity matrices from ground to snow and from snow to snow. Combing the boundary conditions [30], these differential equations could be analyzed in the whole layers. The details of solving these equations could be found in [30,32], and it is demonstrated that the results of the multilayer QCA/DMRT agree well with the CLPX ground measurement [32]. In this paper, the input medium physical parameters of DMRT are: (1) snowpack thickness *t*; (2) physical temperature *T* of the snowpack. From these parameters, the brightness temperature $B_{v}$ in vertical polarization and $B_{h}$ in horizontal polarization at various observation angles could be simulated by the above model.

### 2.2. Deep ConvNet Architecture

In our approach, the ConvNet model is utilized to inverse the thickness *t* and the temperature *T* of the snowpack from its corresponding brightness temperature $B_{v}$ in vertical polarization and $B_{h}$ in horizontal polarization. These arrays of brightness temperature $B_{v}$ in vertical polarization and $B_{h}$ in horizontal polarization can be measured by passive microwave remote sensing based on DMRT numerical simulation. In fact, the brightness temperature $B_{v}$ and $B_{h}$ indicate the feature information of snow [30,32]. In the training process, the thickness *t* and the temperature *T* of the snowpack are utilized as outputs to our deep ConvNets. Their corresponding brightness temperature $B_{v}$ and $B_{h}$ are employed as the inputs in the proposed framework.

Typical ConvNets [24,25] consist of four types of layers: input layers, convolutional layers, pooling layers and fully-connected layers. By stacking these layers together, the proposed ConvNet architecture is formed. As the typical deep neural networks, ConvNet can make use of data in the form of spatially focused images [21,33]. The specific architecture of the proposed ConvNet is presented in Figure 2. Because of the strong capability of ConvNet, our approach can convert the brightness temperature $B_{v}$ in vertical polarization and $B_{h}$ in horizontal polarization with more substantial interference into the corresponding thickness *t* and the temperature *T* of the snowpack.

### 2.3. Data Preparation and ConvNet Training

Considering the requirement for the massive number of input training data is challenging to yield via actual observations, we employ numerical simulation to obtain the input data for the training process of the proposed ConvNet. By PMRS of snowpack based on DMRT numerical simulation, both the brightness temperature $B_{v}$ in vertical polarization and $B_{h}$ in horizontal polarization are measured under incident angle $\theta_{inc}$ evenly distributed within $\left[6^{\circ },75^{\circ }\right]$, and form the ’field-data’ $\left[B_{v},B_{h}\right]$ with the size of 70×2, as is presented in Figure 3. Considering the interference in the actual application scenario, the random noise with signal-to-noise ratio (SNR) up to 0dB is added to the measured the brightness temperature ’field-data’ $\left[B_{v},B_{h}\right]$ to form the input training data, while [t,T] of the snowpack is utilized as outputs to our deep ConvNets.

The procedures of producing training data are formulated into the following two steps: (1) the brightness temperature is firstly calculated from the different thickness *t* and the temperature *T* of the snowpack by DMRT model, where the values of *t* and *T* are within t∈{16cm,18cm,…,44cm} and T∈{250K,252K,…,270K}, respectively. (2) the random noise with signal-to-noise ratio (SNR) up to −10dB is added to the brightness temperature computed in (1) to form one group of input of the proposed ConvNet, while the output is the corresponding *t* and *T*. 5000 groups of the brightness temperature ’field data’ $\left[B_{v},B_{h}\right]$ with noise and the corresponding accurate [t,T] are used as inputs and outputs of the proposed ConvNet. All simulation computation is done on snow-covered ground with the flat bottom surface and with known parameters: grain diameter is 0.1cm, snow density in gm/cc is 0.276, QCA stickiness parameter is 0.1, the ground temperature is 270K and the frequency of measurement microwave is 18.7 GHz.

As shown in Figure 2, the convolutional and activation layer units operate to grasp the input characteristics. In the convolutional layer, we pick the filters (kernel) with a one-dimension (1D) size. Such a 1D kernel has broadly been used in processing text natural language and predicting stock [34,35]. Table 1 presents the number of Convolutional layers and kernel. In addition, the size and the stride of the kernel are detailed. For the output, i.e., the predicted thickness *t* and the temperature *T* of the snowpack, a fully-connected layer is adopted for the prediction, which is fed by the activation layer unit. Herein, the loss function is defined by the half mean squared error between the actual label and the predicted one, i.e., the output of the fully-connected layer. Our proposed approach is benchmarked via the Deep Learning Toolbox in Matlab 2018b [36]. Here, the Adaptive Moment Estimation (Adam) optimizer is selected to optimize the half mean squared error loss function. This is because, compared with other optimizers like Stochastic Gradient Descent (SGD) [37], the Adam optimizer can navigate through the loss surface more successfully. Notably, the learning rate, a hyper-parameter in the proposed framework, can be used to control training error. We set the learning rate as 0.01. In addition, to avoid the over-fitting issue, L2 regularization is used for the improvement of prediction accuracy [38]. All the training is implemented with the full batch.

## 3. Numerical Examples

In this section, the trained Convnet is used to predict the thickness *t* and the temperature *T* of the snowpack. In total, 150 groups of the measured brightness temperature ’field-data’ $\left[B_{v},B_{h}\right]$ with (SNR=0dB) noise calculated by different the thickness and the temperature [t,T] of the snowpack were used as inputs of our trained ConvNet, where the values of *t* and *T* are within t∈{17cm,19cm,…,43cm} and T∈{251K, 253K,…,269K}, respectively.

The comparison of the prediction of *t* and *T* and the actual values is presented in Figure 4a,b respectively. It can be indicated that the predicted data points of both *t* and *T* are closely distributed around the straight line Y=X. Evidently, the proposed ConvNet can effectively realize inversion of the thickness *t* and the temperature *T* of the snowpack even under big interference. According to Figure 4a, the correlation coefficient ($R^{2}$) between the predicted and actual *t* is even 0.9964, while the root mean square error (RMSE) of them is only 0.3869. Thus, despite noise, the proposed ConvNet can predict the thickness *t* under much high accuracy.

Figure 4b shows that the change tendency of the snow temperature *T* in Figure 4b is similar to that of the snow thickness *t* in Figure 4a. From Figure 4b, the correlation coefficient ($R^{2}$) between the predicted *T* and the actual *T* is 0.8380, while the root mean square error (RMSE) of two sets of *T* is 1.7873. The inversion prediction of *T* closed to the scope of trained values has more error than the prediction away from the scope of trained values. This is simply because the predicted values are limited in the scope of trained values. As the values of inverted parameter are close to the end of the scope, those inaccurately predicted values shift toward the other end [21]. In our deep ConvNet approach, the thickness *t* and the temperature *T* can be inverted simultaneously at the nearly equivalent accuracy by the same deep network rather than two separated networks.

Moreover, we compare the ConvNet inversion result of *t* and *T* with that predicted from conventional artificial neural network (ANN) [28], to demonstrate the capability of our deep ConvNet. Conventional ANNs depend on neural unit and hidden layer to fit the relationship between input and output, and have to make use of a large number of neural units to handle the relatively complicated relationship between input and output [28]. The architecture of used ANN has three layers: input, hidden-layer, and output layer. There are 20 hidden-layer units of hyperbolic tangent basis function. The BPNN is implemented using Matlab 2018b with Deep Learning Toolbox [36]. For this case, the increase of ANN hidden layer or its units do not lead to the great improvement of accuracy, but leads to the increase of unexpected computation cost and degrades its efficiency. As shown in Figure 5a,b, the correlation coefficient $\left(R^{2}\right)$ between the predicted *t* and actual *t* is 0.9321 while $R^{2}$ between two sets of *T* is as small as 0.3332. Besides, the root mean square error (RMSE) of the predicted *t* and actual *t* arrives at only 2.1036, and RMSE of two sets of *T* reaches 3.5625. Therefore, the estimation accuracy of the ConvNet is higher than that of the conventional ANN by comparing Figure 4 and Figure 5.

Furthermore, the SVM method is also used to invert the *t* and *T* [21,29]. As is presented in Figure 6a,b, $R^{2}$ between the predicted *t* and actual *t* is 0.9667, while RMSE of them is as small as 0.6232. Besides, $R^{2}$ and RMSE between two sets of *T* are 0.1749 and 6.4291, respectively. Thus, as is shown in Figure 6, the estimation accuracy of the SVM is undoubtedly lower than both the proposed ConvNet and conventional ANN. In addition, both the proposed deep ConvNet and conventional ANN can simultaneously invert the *t* and *T*, while the SVM model has to make use of two different models to undertake the inversion.

As presented in Figure 3, Figure 4, Figure 5 and Figure 6, the proposed deep ConvNet can utilize PMRS data with significant noise to extract the features and retrieve *t* and *T*. The deep ConvNet employs data in the form of spatially focused images to discover recognization and imaging [21,26]. Thus, despite huge noise and abstract PMRS data, the performance of the deep ConvNet could be much better than the other two methods. The overall performance of the three methods is also shown in the Table 2. It is evident that the proposed deep ConvNet can inverse *t* and *T* with the largest $R^{2}$ and the smallest RMSE among the three methods. While $R^{2}$ of the ConvNet method can be about 4.8 times as large as that of SVM for inversing *T*, RMSE of the ConvNet method can be only 18% of that of ANN method for inversing *t*. In this study, we have focused on thickness and temperature, while other parameters could be added to the proposed deep ConvNets. We aim to do this in future studies. Also, extremes of the working ranges of thickness and temperature as well as the lower boundary of SNR in the anti-interference simulation could be included in future research.

## 4. Conclusions

To sum up, a novel inversion method is proposed to extract the layer thickness and temperature of snowpack by using the deep convolutional neural network (ConvNet). The proposed ConvNet consists of three pairs of convolutional and activation layers, following one additional fully connected layer to the inverse of snow parameters. The training of the proposed deep ConvNet is based on simulated data obtained through a dense medium radiative transfer equation (DMRT). The training data also considers the possible interference in a real application scenario. The trained deep ConvNet can inverse the layer thickness and temperature of snowpack within an acceptable accuracy range, which indicates its capacity for anti-interference. Numerical examples indicate the validity of the proposed deep ConvNet for the inversion of parameters of the snowpack. Compared with the conventional artificial neural network and support vector machine, the trained ConvNet can predict the result with higher accuracy. The proposed ConvNet method opens a novel path for deep learning application to passive microwave remote sensing.

## Figures and Tables

**Figure 1 sensors-22-04769-f001:**
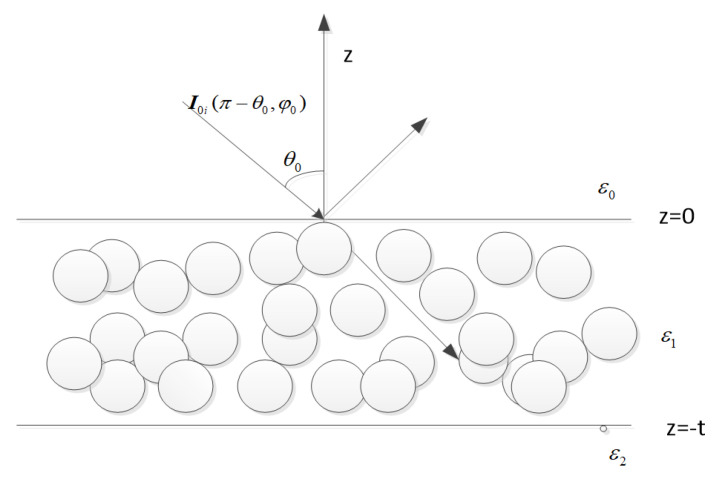
Model microwave emission behavior for multilayer snowpacks.

**Figure 2 sensors-22-04769-f002:**
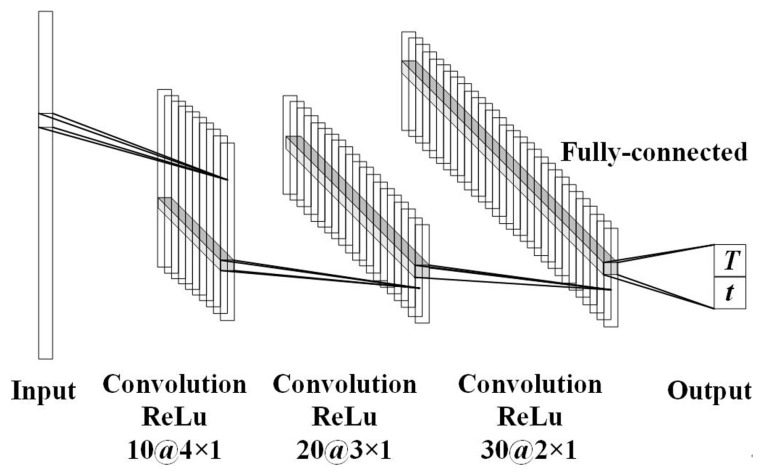
ConvNet architecture for the inversion of snow parameters.

**Figure 3 sensors-22-04769-f003:**
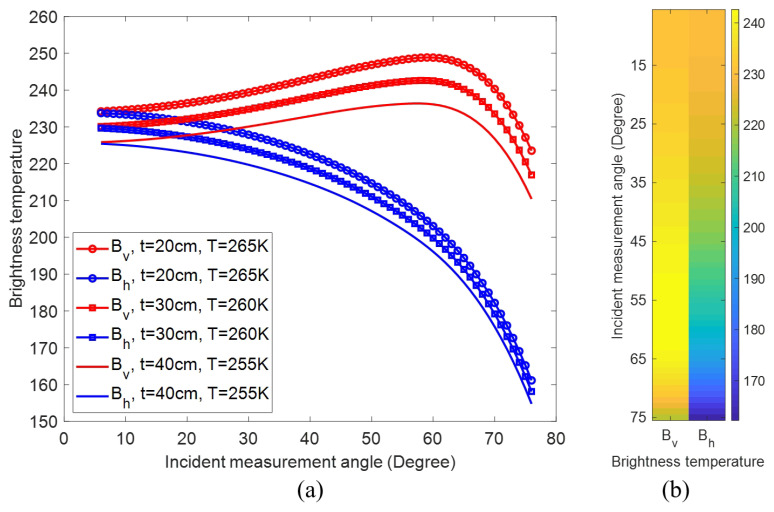
(**a**) the brightness temperature $B_{v}$ in vertical polarization and $B_{h}$ in horizontal polarization with t=30cm and T=260K; (**b**) the ’field-data’ $\left[B_{v},B_{h}\right]$ as the input of ConvNet.

**Figure 4 sensors-22-04769-f004:**
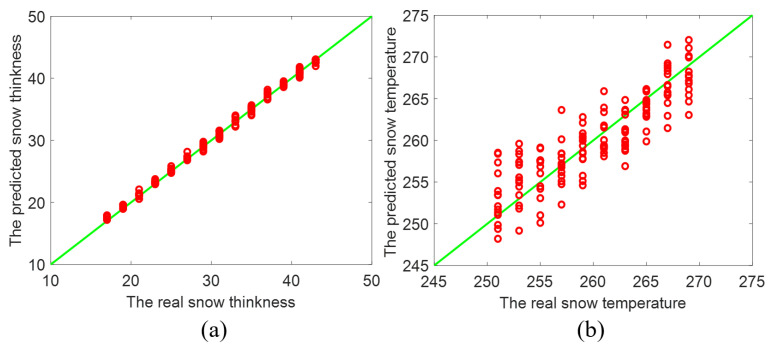
Inverted result of (**a**) the thickness and (**b**) the temperature of the snowpack by the proposed ConvNet method.

**Figure 5 sensors-22-04769-f005:**
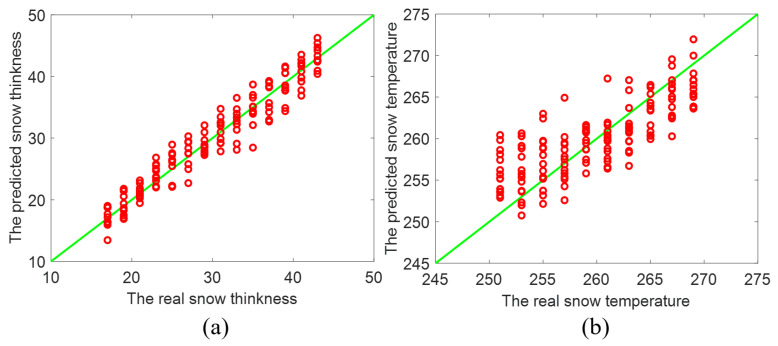
Inverted result of (**a**) the thickness and (**b**) the temperature of the snowpack by the conventional ANN method [14,28].

**Figure 6 sensors-22-04769-f006:**
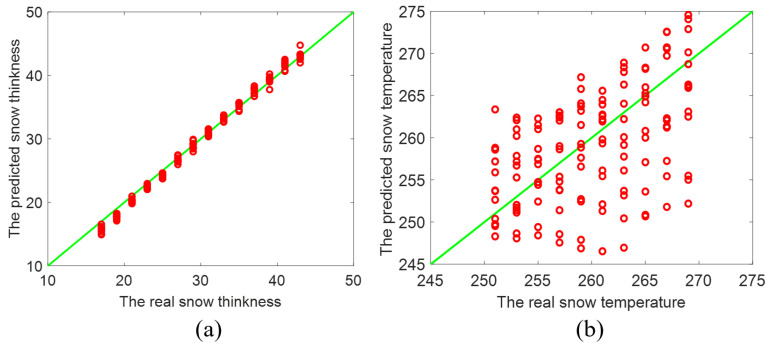
Inverted result of (**a**) the thickness and (**b**) the temperature of the snowpack by the SVM method [21,29].

**Table 1 sensors-22-04769-t001:** ConvNet architecture.

Type	Filter Number	Filter Size	Stride	Input Size	Output Size
Convolution	10	4×1	[2 1]	70×2×1	34×2×10
ReLu				34×2×10	34×2×10
Convolution	20	3×1	[2 1]	34×2×10	16×2×20
ReLu				16×2×20	16×2×20
Convolution	30	2×1	[2 1]	16×2×20	8×2×30
ReLu				8×2×30	8×2×30
Fully-connected Regression				480	2

**Table 2 sensors-22-04769-t002:** Performance Comparison.

		$R^{2}$	RMSE	$\frac{R_{\mathrm{DConvNet}}^{2}}{R^{2}}$	$\frac{{\mathrm{RMSE}}_{\mathrm{DconvNet}\;}}{\mathrm{RSME}}$
	DConvNet	**0.9964**	**0.3869**		
*t*	ANN	0.9321	2.1036	1.0690	0.1839
	SVM	0.9667	0.6232	1.0307	0.6208
	DConvNet	**0.8380**	**1.7873**		
*T*	ANN	0.3332	3.5625	2.5150	0.5017
	SVM	0.1749	6.4291	4.7913	0.2780

## Data Availability

Data are available upon request by email to the corresponding author.

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
