# Peer review of "Snow Parameters Inversion from Passive Microwave Remote Sensing Measurements by Deep Convolutional Neural Networks"

_sensors, 2022, doi:10.3390/s22134769_

Round 1
Reviewer 1 Report
Excellent work. Congratulations
1. As we know, snow is not one and the same media, its condition depends on many factors: outside temperature, age of the layer, humidity, snowfall, etc. Authors did not mention, how their method may be sensitive to all these conditions. Should it be different programs for every snow condition?
2. Most difficult may be the case when there are many different layers: ice, old snow, new snow - and how is it possible to train the system, if the condition of this snow is changing practically every hour?
3. It is not clear how and where presented experimental data have been collected and how the comparison was made.
4. Interesting, whether it is possible to use this technology during sports ski competitions? Predicting snow conditions would be extremely important for professionals in this field. 5. Based on the presented approach would it be possible to create a classification of different types of snow?
Author Response
Please refer to the response to Reviewer 1 in the attachment.

Reviewer 2 Report
This paper gives a substantial investigation on the snow parameters inversion from passive microwave remote sensing measurements by deep convolutional neural networks. The considered topic is interesting and useful. The theoretical derivation is rigorous and the results appear to be correct and believable. In general, the manuscript is well-written (although I gave a good number of corrections and suggestions) and provides an interesting conclusion. Nevertheless, the reviewer holds some concerns about this work which you can find below. I suggest that the authors revise and improve the manuscript accordingly.
In fact, there is a long list of papers considering a similar system. What are their achievements? What is the gap existed? The considered DCNN has been well investigated in many other works before. Please describe the contribution of this work more clearly. What are the main novelties of the present work with respect to similar published papers in the literature? I suggest that you compare your work with the works in the literature in a table. In addition, the major contributions of this work should be written more solid.
More recent works in wave transmission in complex media should be discussed in the introduction section of the revised manuscript.
The following reference could help the readers and prevent from misleading of previously undertaken research works.
Terahertz waves propagation in an inhomogeneous plasma layer using the improved scattering-matrix method
Speed and attenuation of acoustic waves in snow: Laboratory experiments and modeling with Biot's theory
The mathematical analysis in theoretical section should combine with the physical meaning of the system model. Otherwise, the mathematical analysis is general in any similar scenario.
The authors should justify their choice of parameters using suitable references.
The practical implementation of the proposed work along with the scope for future works should be discussed in the manuscript.
Author Response
Please refer to the response to Reviewer 2 in the attachment.

Round 2
Reviewer 2 Report
This manuscript can be accepted this time since all my concerns have been solved.
Author Response
Thanks for your recommendation.